# Mapping the Implementation of a Clinical Pharmacist-Driven Antimicrobial Stewardship Programme at a Tertiary Care Centre in South India

**DOI:** 10.3390/antibiotics10020220

**Published:** 2021-02-23

**Authors:** Vrinda Nampoothiri, Akkulath Sangita Sudhir, Mariam Varsha Joseph, Zubair Mohamed, Vidya Menon, Esmita Charani, Sanjeev Singh

**Affiliations:** 1Department of Infection Control and Epidemiology, Amrita Institute of Medical Science, Amrita Vishwa Vidyapeetham, Kochi 682041, Kerala, India; vrindan@aims.amrita.edu (V.N.); sangitas@aims.amrita.edu (A.S.S.); mariamvj@aims.amrita.edu (M.V.J.); 2Anesthesiology and Critical Care Medicine, Amrita Institute of Medical Science, Amrita Vishwa Vidyapeetham, Kochi 682041, Kerala, India; zubairum22623@aims.amrita.edu; 3Department of Medicine, Division of Infectious Diseases, Amrita Institute of Medical Science, Amrita Vishwa Vidyapeetham, Kochi 682041, Kerala, India; vidyamenon@aims.amrita.edu; 4Health Protection Research Unit in Healthcare Associated Infections and Antimicrobial Resistance, Imperial College, London W12 0NN, UK; e.charani@imperial.ac.uk

**Keywords:** antimicrobial stewardship, clinical pharmacist, antimicrobial resistance, antimicrobial management, defined daily dose

## Abstract

In many parts of the world, including in India, pharmacist roles in antimicrobial stewardship (AMS) programmes remain unexplored. We describe the evolution and effect of the role of adding clinical pharmacists to a multidisciplinary AMS at a tertiary care teaching hospital in Kerala, India. Through effective leadership, multidisciplinary AMS (February 2016) and antitubercular therapy (ATT) stewardship programmes (June 2017) were established. Clinical pharmacists were introduced as core members of the programmes, responsible for the operational delivery of key stewardship interventions. Pharmacy-led audit and feedback monitored the appropriateness of antimicrobial prescriptions and compliance to AMS/ATT recommendations. Between February 2016 and January 2017, 56% (742/1326) of antimicrobial prescriptions were appropriate, and 54% (318/584) of recommendations showed compliance. By the third year of the AMS, appropriateness increased to 80% (1752/2190), and compliance to the AMS recommendations to 70% (227/325). The appropriateness of ATT prescriptions increased from a baseline of 61% (95/157) in the first year, to 72% (62/86, June 2018–February 2019). The compliance to ATT recommendations increased from 42% (25/60) to 58% (14/24). Such a model can be effective in implementing sustainable change in low- and middle-income countries (LMICs) such as India, where the shortage of infectious disease physicians is a major impediment to the implementation and sustainability of AMS programmes.

## 1. Introduction

Antimicrobials constitute around one third of the health budgets in developing countries, where the burden of infectious diseases is immense [1]. A study conducted on the use of antibiotics in 76 countries over 16 years reported that antibiotic consumption rates in low- and middle-income countries (LMICs) are on par to rates observed in high-income countries [2]. The overuse of antibiotics is leading to an increase in antimicrobial resistance [3]. Evidence supports the effectiveness of antimicrobial stewardship (AMS) programmes to optimise care, and preserve the efficacy of antibiotics for current and future generations [4,5]. AMS refers to “coordinated interventions designed to improve and measure the appropriate use of antimicrobial agents by promoting the selection of the optimal antimicrobial drug regimen, dose, duration of therapy and route of administration” [6].

Implementation of an effective AMS programme requires a multidisciplinary approach, involving a variety of experts including an infectious diseases physician, a clinical pharmacist with infectious diseases training, a clinical microbiologist, an information system specialist, a hospital epidemiologist, and an infection control specialist [7]. The majority of guidelines available around the world advocate the role of the clinical pharmacist to be integral to AMS teams [8,9,10]. The American Society of Health-System Pharmacists recognises that pharmacists have a responsibility to take prominent roles in AMS and participate in the infection prevention and control programmes of healthcare organisations [11]. Clinical pharmacists have expert knowledge of drugs which can be utilised to rationalise antibiotic use and prevent the emergence of resistance. In countries such as the United Kingdom, a specialised antibiotic pharmacists’ role in AMS includes providing expert advice on the use of antibiotic management, education on antibiotic prescribing for health care professionals, to act as a liaison between pharmacy and microbiology departments, formulary enforcement, developing and maintaining antibiotic guidelines, monitoring and feedback of antibiotic use, co-developing antibiotic policy, and antibiotic serum level monitoring and dosing [12]. Similarly, in the United States, the role of clinical pharmacists have expanded over time, and they are considered core members of the AMS team and assist in appropriate antibiotic utilization [13].

Clinical pharmacy was introduced to pharmacy education in India with a postgraduate diploma course in 1996. This was followed by a postgraduate programme (MPharm) in 1997 and then a Doctor of Pharmacy (Pharm D) programme in 2008 [14]. Even though this specialisation of pharmacy has been established in India for quite some time now, there are still not enough opportunities for clinical pharmacists in the hospital setting. Whilst clinical pharmacists have slowly started being integrated into private hospital settings in India, their role remains limited to medicines reconciliation, prevention, identification and reporting of adverse drug reactions, detection and prevention of drug interactions, and providing support to the physicians and surgeons during ward rounds. Opportunities for clinical pharmacist services in the public hospitals in India remain rare. Furthermore, AMS programmes remain underdeveloped in India, with very few hospitals (public and private) having clinical pharmacists working in AMS. A survey conducted in 2013 to investigate the practice of AMS among 20 hospitals in India highlighted the absence of infectious diseases physicians and clinical pharmacists as major gaps which need to be addressed [15]. We describe how a multidisciplinary AMS programme in a teaching hospital in South India evolved to be driven by clinical pharmacists, and in the process created new opportunities for clinical pharmacy. We also elaborate on the challenges faced while implementing this model, and the measures that were taken to overcome them. 

## 2. Materials and Methods

### 2.1. Ethics

Institutional Ethics approval was not required prior to the initiation of the programme as this was a quality improvement initiative.

### 2.2. Setting

This work was conducted at Amrita Institute of Medical Sciences (AIMS), Kochi, Kerala, India, which is a 1300-bedded tertiary care teaching hospital. The hospital started the Doctor of Pharmacy (Pharm D) programme, both regular and post baccalaureate, in the year 2010. Each year, a total of 40 students graduate with this degree from the academic institution affiliated with the hospital. 

### 2.3. Implementation and Development of A Clinical Pharmacist Driven AMS Programme

An AMS programme was initiated in the hospital in February 2016. A multidisciplinary team was commissioned by the hospital Medical Superintendent (S. Singh) and included a physician (V.M.), microbiologist, and intensivist (Z.M.) [16,17,18]. The team included clinicians with experience in different countries, who, as a result, were familiar with the potential role for pharmacists in AMS. During this period, there was a lack of clinical pharmacists trained in AMS in Kerala. One of the initial initiatives of the AMS team was to train and create a role for Doctor of Pharmacy (Pharm D) interns, who were pursuing their course in the pharmacy college within the hospital campus. The clinical pharmacy department, under the pharmacy college in the hospital campus, selects three Pharm D interns and provides them a clinical posting in AMS for a period of 2 months on a rotational basis. These interns report directly to the medical superintendent during their 2-month posting. A bespoke induction training programme was created for the Pharm D interns by the clinicians in the AMS committee (Box 1). End of rotation feedback is provided on their progress, and they are given a certificate of completion from the AMS team.

Box 1Induction training of clinical pharmacist by the clinicians in the antimicrobial stewardship (AMS) team.Induction training for Pharm D interns include:
Objectives of the AMS programme;Roles and responsibilities of the pharmacist in AMS;Basic mechanisms of antimicrobials including synergistic antibiotics, double anaerobic cover combinations;Assessing the appropriateness of prescription using five R’s: Right indications, Right drug, Right dose, Right frequency and Right duration;Interpreting microbiological culture reports: distinguishing between true infection and colonization and/or contamination.


A case report form (CRF) adapted from guidelines of the Society of Hospital Epidemiologist, Infectious Diseases Society of America and Centers for Disease Control and Prevention, was developed by the clinicians to guide the interns to collect the patient data deemed necessary for evaluating the antimicrobial prescribing on the wards [10,16,19,20]. The CRF (available as Appendix A) was piloted and later modified to include patient demographics, details of microbiology cultures and sensitivities, ventilator parameters, laboratory parameters, details of invasive devices, dose, frequency, and indication of antimicrobials prescribed throughout the patient’s hospital stay. The doctors in the team provided rigorous training to the Pharm D interns on how to collect the patient data, how to assess appropriateness of antimicrobial therapy, and how to formulate the recommendations. Routine training was provided on a case-to-case basis during the daily review meeting with the AMS committee. A recommendation form was also created by the AMS team to communicate their recommendations to the primary team of the patient. The recommendation form (available as Appendix A) also had basic details of the patient, details of relevant cultures, antimicrobials prescribed, and the recommendation(s) from the AMS team, and this was to be filed in the patient file. 

The AMS team initially audited three reserved antimicrobials, based on the consumption data, namely colistin, polymyxin B and tigecycline, which over time expanded to a total of 14 reserved antimicrobials. The workflow of the AMS team has been outlined in Figure 1.

The key components of the AMS programme included:A multidisciplinary team, including administrator, physician, intensivist, microbiologist and clinical pharmacist.Well established roles and responsibilities for each stakeholder in the AMS team.Reserve antimicrobial list based on antibiogram.Daily AMS meetings to review appropriateness of reserved antimicrobial prescriptions.Timely feedback to the primary team on a case-to-case basis.Development and dissemination of institutional antibiotic guidelines.Structured training for pharmacist and pharmacy interns.

Within six months of AMS implementation, in order to maintain consistency in the data collection and storage, the administrative champion in the team decided to appoint a full-time clinical pharmacist (V.N.) who had participated in AMS activities as part of their internship posting. The interns continued to be posted in AMS, but the newly appointed clinical pharmacist was given the responsibility of giving initial induction to the interns, and played a key role in routine training of the interns, along with the clinicians. The capacity of the clinical pharmacist was gradually built to take up more responsibilities in the programme (Figure 2). The additional responsibilities included quality improvement initiatives, such as auditing the medication reconciliation, and inpatient and outpatient prescription auditing across the hospital conducted by Pharm D students and interns under the guidance of the AMS clinical pharmacists. 

In addition to auditing the reserved antimicrobials in the hospital, the AMS team also developed dosing protocols for reserved antibiotics such as colistin. These protocols were based on international guidelines due to the non-availability of local data. Recognising the gap in local population data for evaluating pharmacokinetic and pharmacodynamics parameters of antimicrobials, including colistin, the AMS team decided to broaden the responsibilities of the clinical pharmacist and appointed an additional clinical pharmacist (S Sudhir) to the AMS team. This allowed for the gathering of local population data to develop bespoke dosing schedules for antimicrobials such as colistin, allowing for more population-targeted guidelines [21]. The AMS team, led by the clinical pharmacists and approved by the microbiologist, expanded on the existing hospital antibiogram to develop department and body fluid-specific antibiograms. 

The evolving profile of the AMS team led to appointment of more clinical pharmacists (MVJ) to support the work, including a separate antitubercular therapy (ATT) stewardship which was initiated in June 2017 [22]. ATT stewardship began by auditing the hospitalised patients on ATT and has since expanded to monitoring outpatient prescribing of antituberculosis medications and the follow-up and clinical management of the patients. These initiatives were funded by the hospital, with the exception of ATT stewardship, which was funded by Revised National Tuberculosis Control Programme.

The clinical pharmacists in the programme were also given opportunities to work with national and international experts in AMS through various collaborative projects. This gave them opportunities for capacity building, training Pharm D interns, nurses, and pharmacists in other hospitals and colleges by speaking at various national seminars which improved their teaching and training skills, leading to a “train the trainer" model. Clinical pharmacists were also encouraged to write research proposals for grants, contribute to research articles, and present their work at various national and international conferences. 

## 3. Results

At the end of 1 year of implementation, AMS team had audited 1326 prescriptions of which 742 (56%) were found to be appropriate. Recommendations were filed for 584 inappropriate prescriptions of which 318 (54%) were complied with. By the third year, the AMS team was able to increase the appropriateness of prescriptions to 80% (1752/2190) and compliance to recommendations to 70% (227/325).

The ATT stewardship team reviewed 157 prescriptions in the first year of its implementation, of which 61% (95/157) were appropriate, and 42% (25/60) of recommendations showed compliance. From June 2018 to February 2019, 72% (62/86) ATT prescriptions were appropriate and 58% (14/24) were compliant with ATT recommendations.

Table 1 shows the reduction in consumption of certain reserved antimicrobials being audited by the AMS team. The consumption of antimicrobials has been expressed in terms of Defined Daily Dose (DDD)/1000 patient days. When AMS was initiated in the hospital, they identified many instances where the duration of antimicrobial therapy extended beyond the recommended duration based on the focus of infection. One of the key initiatives undertaken by the clinical pharmacist in the AMS was the routine follow-up of all reserved antimicrobial prescriptions on the 7th day and 14th day. This provided the opportunity to intervene to deescalate or discontinue the antimicrobial, thereby resulting in a reduction in the antimicrobial consumption. Even though the AMS model in the hospital was well structured with defined roles for clinical pharmacists, they faced various challenges, especially in the initial years of AMS implementation. Practical solutions were developed to overcome the challenges faced by clinical pharmacists in the programme, as described in Table 2. 

## 4. Discussion

In this paper, we have mapped the clinician-led and clinical pharmacist-driven AMS, demonstrating that it can be effective in implementing sustainable change in LMICs such as India, where the shortage of infectious diseases physicians is a major impediment. The multidisciplinary model in this hospital has the potential for scale up to other hospitals in India and LMICs.

Pharmacist-driven AMS models have been shown to produce positive results in other LMICs settings too. An implementation study on a pharmacist-driven AMS programme across 47 hospitals in South Africa showed a significant reduction in inappropriate antibiotic use [23]. Pharmacist-led antimicrobial audit and feedback conducted in a referral hospital in Ethiopia also showed reduction in antimicrobial consumption [24]. 

Even though the Pharm D programme was introduced in India over a decade ago, the clinical pharmacists are still facing various challenges to gain recognition [14]. In spite of having policy documents from India suggesting that clinical pharmacists are key professionals in AMS programme [25], not many hospitals have an established role for them in AMS. In contrast, our AMS had clinical pharmacists with specific roles and responsibilities from the beginning, which expanded over time. This model has shown to increase the appropriateness in the prescription of reserved antimicrobials and antitubercular therapy, in addition to a reduction in the consumption of certain reserved antimicrobials in the hospital. Such a model of AMS can also create more opportunities for clinical pharmacists. 

The clinical pharmacist in the team also trained pharmacy interns and pharmacists from other hospitals on AMS. This kind of “training the trainer” model provided an opportunity for pharmacists to better understand their role in AMS and resulted in the dissemination to other hospitals and standardisation of a clinical pharmacist-driven model. This also enabled the clinical pharmacist in our programme to improve their assigned roles by learning from other hospitals. A study conducted in China showed that a “train the trainer” programme, conducted among pharmacists on pharmacy clinical services in a tertiary care centre, enabled an increase in clinical pharmacy consultations by clinical pharmacists, who were able to expand their services beyond consultations [26].

Having a consistent administrative backup was key to the establishment and acceptance of this AMS model. The strong support of the doctors in the AMS also proved crucial in establishing the role of clinical pharmacist. Strong leadership support is essential in fostering acceptance and “buy-in” for pharmacists being integrated in AMS teams [27]. This is critical to the success of integrating interdisciplinarity in AMS. 

A narrative review conducted on enhancing pharmacists’ roles in developing countries to overcome the challenge faced by AMS, suggests that the major barriers to the delivery of comprehensive pharmacy services include the shortage of pharmacists, the lack of pharmaceutical care training programmes, and institutional obstacles [28]. The constant and continuous efforts taken by the AMS team helped in improving the acceptance and the scope of the programme, as well as the clinical pharmacists’ skills and capability. 

Pharmacists are also said to have a key role to play in diagnostic stewardship [29], outpatient antibiotic stewardship [30] and infection prevention and control activities in hospitals [31]. These are some areas that are being explored by clinical pharmacists in our AMS to further widen their role in the hospital. 

## 5. Conclusions

Clinical pharmacists have a critical role in AMS, and can be effective in implementing sustainable change in LMICs such as India, where the shortage of infectious diseases physicians is a major impediment to driving change. A multidisciplinary approach to AMS has the potential for scale up to other hospitals in India and LMICs, where professional boundaries still limit the integration of pharmacists in clinical services. 

## Figures and Tables

**Figure 1 antibiotics-10-00220-f001:**
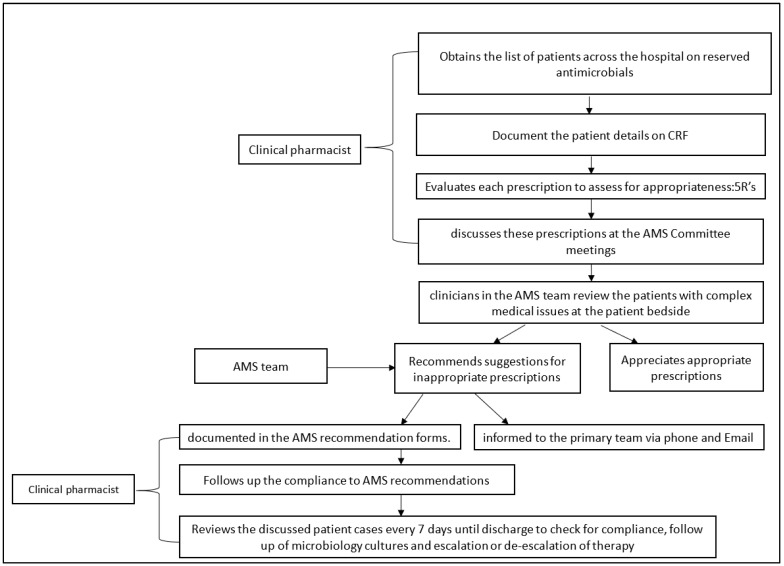
Workflow of AMS team at Amrita Institute of Medical Sciences (AIMS), Kochi.

**Figure 2 antibiotics-10-00220-f002:**
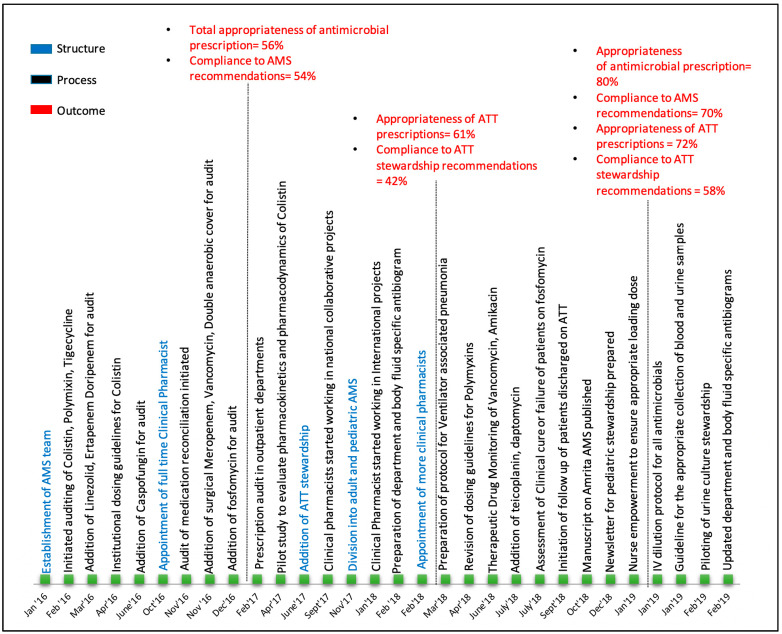
Evolution of clinical pharmacists’ role in AMS at AIMS, Kochi.

**Table 1 antibiotics-10-00220-t001:** Decrease in consumption of reserved antimicrobials, in terms of Daily Defined Dose (DDD) for 1000 patient days, post AMS at Amrita Hospital, Kochi.

Drug	Post-AMS Consumption DDD for 1000 Patient Days		
	February 2016–January 2017	February 2017–January 2018	February 2018–January 2019
Colistin	25.18	25.8	11.7
Amphotericin B	12.01	10.08	0.3
Caspofungin	0.69	0.74	0.3
Ertapenem	1.72	1.17	1.1
Linezolid	36.13	22.54	17.9
Meropenem	71.22	69.58	52.9
Anidulafungin	0.74	0.43	0
Tigecycline	5.17	4.11	3.2

**Table 2 antibiotics-10-00220-t002:** Challenges faced by clinical pharmacist in AMS at AIMS, Kochi, and the measures undertaken to overcome them.

Challenges Faced	Measures Undertaken to Overcome the Challenges
Clinical pharmacists had the theoretical knowledge about antimicrobials, its mechanisms, and doses in various indications, but they did not have the clinical expertise to justify the appropriateness of a prescription.	The clinicians in the AMS team trained the clinical pharmacist, on a case-to-case basis, in assessing the appropriateness of prescription using 5Rs. They were given opportunities to attend various national conferences to improve their knowledge in antimicrobial therapy. They were also encouraged to attend Massive Open Online Courses (MOOC) on AMS. Weekly journal clubs led by the clinical pharmacists to provide shared learning on AMS.
Initial resistance to, and criticism of, the AMS programme from the primary teams, including what they considered to be a “closed room approach”, where the clinical pharmacists were sitting inside a room and making recommendations to modify the antimicrobials without actually seeing the patient and assessing them clinically.	The administrative champion clarified to the primary teams that the AMS model was a clinician-led, clinical pharmacist-driven approach. To support AMS recommendations, the clinical pharmacists provided the evidence base for their recommendations. These references would include the national or international guidelines from learned societies or journal articles, etc. The departments frequently non-compliant to the recommendations were called in for a meeting with the administrative champion in the AMS team. This was a platform for the AMS team to explain their rationale, and for the doctors to discuss their justifications for non-compliance. This was an iterative process and helped team building, mitigating the initial skepticism that characterised the clinicians’ response to the AMS programme. Initially the clinical pharmacists communicated the recommendations to either the staff nurse or the junior doctors in the team. On identifying that these were not being communicated effectively to the senior clinicians in the team, who are the key decision makers, clinical pharmacists started directly informing the senior consultants whenever they were available. Clinical pharmacists referred complex cases to clinicians in the AMS team, who reviewed the patients and discussed the recommendations with the primary team.
Due to the presence of a limited number of clinical pharmacists in AMS, the programme was not able to expand the list of antimicrobials being audited beyond reserved antimicrobials. Hence, it was unclear whether the reduction in the consumption of reserved antimicrobials had caused an increase in the consumption of first-line or second-line agents.	The team initiated point prevalence studies at specific intervals, wherein, on a particular date, consumption of all antibiotics across the hospital was collected by the clinical pharmacist. This gave the team an idea about the consumption of the non-reserved antimicrobials in the hospital.

## Data Availability

Results presented in this study are from the data routinely collected by the AMS team at the setting. These are confidential data to which only members of the AMS team has access to. Upon reasonable request, anonymized data can be made available.

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
