# Peer review of "Mapping the Implementation of a Clinical Pharmacist-Driven Antimicrobial Stewardship Programme at a Tertiary Care Centre in South India"

_antibiotics, 2021, doi:10.3390/antibiotics10020220_

Round 1
Reviewer 1 Report
Well written manuscript on an important topic, in an area characterized by high antimicrobial consumption. Methods are sound, results directly related to the research topic and conclusions derived from study findings.
Author Response
Reviewer 1 |
Comment |
Well written manuscript on an important topic, in an area characterized by high antimicrobial consumption. Methods are sound, results directly related to the research topic and conclusions derived from study findings. |
Thank you for highlighting the strengths of this work. |
Please see the detailed cover letter attached.

Reviewer 2 Report
Great article describing quality improvement initiative with great results. Congrats!
i would not call it a study and also would use generic names for all drugs so if anyone one decide to implement can easily follow.
Author Response
Reviewer 2 |
Comment |
Great article describing quality improvement initiative with great results. Congrats! i would not call it a study and also would use generic names for all drugs so if anyone one decide to implement can easily follow.
|
We have removed the word ‘study’ in relation to how we describe this work.
We have only used generic names for the drugs mentioned in this manuscript. |
Please see the detailed cover letter attached.

Reviewer 3 Report
In this report, the authors described the evolution and effect of the role of adding clinical pharmacists to a multidisciplinary antimicrobial stewardship (AMS) programme at a tertiary care teaching hospital in India. Following are my questions and comments to serve as a starting point for reworking the manuscript.
1. When taking into account the authors’ previous publication (Ref 15, Open Forum Infect Dis. 2019, 6(4): ofy290, doi: 10.1093/ofid/ofy290), a high similarity was found between them. So please justify the difference and highlight the advancement of this study.
2. Title: Please change the “India” to “south India” since the study was conducted in Kochi, Kerala.
3. Please avoid use acronym if it was mentioned only once in the article. For example, Antimicrobial Resistance (AMR) in Line 44; Revised National Tuberculosis Control Programme (RNTCP) in Line 162, etc.
4. Line 64: “In contrast, in countries like the United Kingdom, specialised 64 antibiotic pharmacists have a key role to play in AMS.” Any supporting references?
5. The introduction should be intensively improved because it routinely introduces some background. It is hard for the readership to understand the significance of this work from the current introduction.
6. Figure 1 has been reported in Ref 15 and thus cannot be reused here.
7. The quality of Figure 2 is too poor. I recommend replacing it with high-quality one.
Author Response
Reviewer 3 |
Comment |
In this report, the authors described the evolution and effect of the role of adding clinical pharmacists to a multidisciplinary antimicrobial stewardship (AMS) programme at a tertiary care teaching hospital in India. Following are my questions and comments to serve as a starting point for reworking the manuscript.
|
Our previous publication was aimed at demonstrating the impact of an Antimicrobial Stewardship(AMS) program in South India. In that we have elaborated the implementation process and the outcomes of the model we implemented. We have mentioned that clinical pharmacists are an integral part of the AMS team. In the current publication submitted to the Antibiotics Journal special issue, we have highlighted how the role of the clinical pharmacist in the program has evolved in the three years post AMS implementation. Our focus is to bring out that AMS is a platform where clinical pharmacists have a great potential as is evident from other countries like UK and USA where their role in this program was initially established. We have shown how the role of clinical pharmacists in the program expanded gradually even outside the program to be involved in various national and international projects, training of pharmacist and pharmacy students on their role in AMS and key roles in various quality improvement projects within the hospital. We have also discussed the challenges we faced and the steps taken to overcome the same. We have tried to show that AMS is a new arena where clinical pharmacist can be utilised in India where there are lack of opportunities for clinical pharmacist in a hospital setting. The intention behind elaborating every step was so that it would be beneficial for any other institution who would like to implement a similar model that could provide more opportunities for clinical pharmacists in India.
|
2. Title: Please change the “India” to “south India” since the study was conducted in Kochi, Kerala. |
The title has been changed as per recommendation. |
3. Please avoid use acronym if it was mentioned only once in the article. For example, Antimicrobial Resistance (AMR) in Line 44; Revised National Tuberculosis Control Programme (RNTCP) in Line 162, etc. |
Thank you for pointing this out. We have removed the acronyms from the specified lines. |
4. Line 64: “In contrast, in countries like the United Kingdom, specialised 64 antibiotic pharmacists have a key role to play in AMS.” Any supporting references? |
We have elaborated the roles of these pharmacist in the next sentence and have quoted the reference there (ref no:12). We have now reworded the sentence and have quoted this reference. |
5. The introduction should be intensively improved because it routinely introduces some background. It is hard for the readership to understand the significance of this work from the current introduction. |
Thank you for your suggestion. We have modified the introduction to better bring out the significance of this work. |
6. Figure 1 has been reported in Ref 15 and thus cannot be reused here. |
A figure similar to figure 1 was used in our previous publication. We further edited the previously published figure for this paper to specifically focus on the role played by the clinical pharmacist in our AMS. |
7. The quality of Figure 2 is too poor. I recommend replacing it with high-quality one. |
Thank you for your input. We have tried to further improve the quality of figure 2. |
Please the detailed cover letter attached.

Round 2
Reviewer 3 Report
The authors have addressed my concerns in this revision.